# SimFBO: Towards Simple, Flexible and Communication-efficient Federated Bilevel Learning

**Yifan Yang, Peiyao Xiao and Kaiyi Ji**
Department of Computer Science and Engineering
University at Buffalo
Buffalo, NY 14260
{yyang99, peiyaoxi, kaiyiji}@buffalo.edu

## Abstract

Federated bilevel optimization (FBO) has shown great potential recently in machine learning and edge computing due to the emerging nested optimization structure in meta-learning, fine-tuning, hyperparameter tuning, etc. However, existing FBO algorithms often involve complicated computations and require multiple sub-loops per iteration, each of which contains a number of communication rounds. In this paper, we propose a simple and flexible FBO framework named SimFBO, which is easy to implement without sub-loops, and includes a generalized server-side aggregation and update for improving communication efficiency. We further propose System-level heterogeneity robust FBO (ShroFBO) as a variant of SimFBO with stronger resilience to heterogeneous local computation. We show that SimFBO and ShroFBO provably achieve a linear convergence speedup with partial client participation and client sampling without replacement, as well as improved sample and communication complexities. Experiments demonstrate the effectiveness of the proposed methods over existing FBO algorithms.

## 1 Introduction

Recent years have witnessed significant progress in a variety of emerging areas including meta-learning and fine-tuning [11, 52], automated hyperparameter optimization [13, 10], reinforcement learning [31, 21], fair batch selection in machine learning [54], adversarial learning [76, 40], AI-aware communication networks [27], fairness-aware federated learning [75], etc. These problems share a common nested optimization structure, and have inspired intensive study on the theory and algorithmic development of bilevel optimization. Prior efforts have been taken mainly on the single-machine scenario. However, in modern machine learning applications, data privacy has emerged as a critical concern in centralized training, and the data often exhibit an inherently distributed nature [70]. This highlights the importance of recent research and attention on federated bilevel optimization, and has inspired many emerging applications including but not limited to federated meta-learning [9], hyperparameter tuning for federated learning [25], resource allocation over communication networks [27] and graph-aided federated learning [71], adversarial robustness on edge computing [46], etc. In general, the federated bilevel optimization problem takes the following mathematical formulation.

$$\min_{x \in \mathbb{R}^p} \Phi(x) = F\big(x, y^*(x)\big) := \sum_{i=1}^{n} p_i f_i(x, y^*(x)) = \sum_{i=1}^{n} p_i \mathbb{E}_\xi \Big[ f_i\big(x, y^*(x); \xi_i\big) \Big]$$

$$\text{s.t. } y^*(x) = \arg\min_{y \in \mathbb{R}^q} G(x, y) := \sum_{i=1}^{n} p_i g_i(x, y) = \sum_{i=1}^{n} p_i \mathbb{E}_\zeta \big[ g_i(x, y; \zeta_i) \big] \quad (1)$$

37th Conference on Neural Information Processing Systems (NeurIPS 2023).

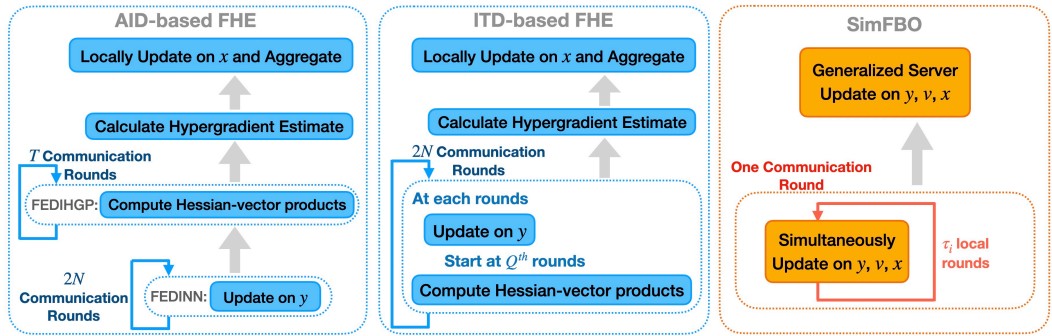

Figure 1: Comparison of AID-based federated hypergradient estimation (FHE) in FedNest [65] (left), ITD-based FHE in AggITD [69] (middle) and our proposed SimFBO (right) at each iteration.

where $n$ is the total number of clients, the outer- and inner-functions $f_i(x, y)$ and $g_i(x, y)$ for each client $i$ take the expectation forms w.r.t. the random variables $\xi_i$ and $\zeta_i$, and are jointly continuously differentiable. However, efficiently solving the federated problem in eq. (1) suffers from several main challenges posed by the federated hypergradient (i.e., $\nabla\Phi(x)$) computation that contains the second-order global Hessian-inverse matrix, the lower- and upper-level data and system-level heterogeneity, and the nested optimization structure. To address these issues, [25, 65, 24, 22] proposed approximate implicit differentiation (AID)-based federated bilevel algorithms, which applied the idea of non-federated AID-based estimate in [15] to the federated setting, and involve two sub-loops for estimating the global lower-level solution $y^*(x)$ and the Hessian-inverse-vector product, respectively. [69] then proposed AggITD by leveraging the idea of iterative differentiation, which improved the communication efficiency of AID-based approaches by synthesizing the lower-level optimization and the hypergradient computation into the same communication sub-loop. However, some limitations still remain in these approaches.

- First, the sub-loops, each with a large number of communication rounds, often compute products of a series of matrix-vector products, and hence can complicate the implementation and increase the communication cost.

- Second, the practical client sampling **without** replacement has not been studied in these methods due to challenges posed by the nested structure of AID- and ITD-based federated hypergradient estimators.

- Third, as observed in the single-level federated learning [67], in the presence of heterogeneous system capabilities such as diverse computing power and storage, clients can take a variable number of local updates or use different local optimizers, which may make these FBO algorithms converge to the stationary point of a different objective.

## 1.1 Our Contributions

In this paper, we propose a communication-efficient federated bilevel method named SimFBO, which is simple to implement without sub-loops, flexible with a generalized server-side update, and resilient to system-level heterogeneity. Our specific contributions are summarized below.

**A simple and flexible implementation.** As illustrated in Figure 1, differently from AID- and ITD-based approaches that contain multiple sub-loops of communication rounds at each iteration, our proposed SimFBO is simpler with a single communication round per iteration, in which three variables $y, x$ and $v$ are updated simultaneously for optimizing the lower- and upper-level problems, and approximating the Hessian-inverse-vector product. SimFBO also includes a generalized server-side update on $x, y, v$, which accommodates the client sampling without replacement, and allows for a flexible aggregation to improve the communication efficiency.

**Resilient server-side updates to system-level heterogeneity.** In the presence of heterogeneous local computation, we show that the naive server-side aggregation can lead to the convergence to a stationary point of a different objective. To this end, we propose System-level heterogeneity robust FBO (ShroFBO) building on a normalized version of the generalized server-side update with correction, which provably converges to a stationary point of the original objective.

| Algorithm | System-level heterogeneity | Partial participation | Without replacement | Linear speedup | Samples complexity | Communication complexity |
|---|---|---|---|---|---|---|
| FedNest [65] | ✗ | ✗ | ✗ | ✗ | $\mathcal{O}(\epsilon^{-2})$ | $\mathcal{O}(\epsilon^{-2})$ |
| FBO-AggITD [69] | ✗ | ✗ | ✗ | ✗ | $\mathcal{O}(\epsilon^{-2})$ | $\mathcal{O}(\epsilon^{-2})$ |
| FedBiO [36] | ✗ | ✗ | ✗ | ✓ | $\mathcal{O}(\epsilon^{-2.5}n^{-1})$ | $\mathcal{O}(\epsilon^{-1.5})$ |
| FedMBO [24] | ✗ | ✓ | ✗ | ✓ | $\mathcal{O}(\epsilon^{-2}P^{-1})$ | $\mathcal{O}(\epsilon^{-2})$ |
| SimFBO (this paper) | ✗ | ✓ | ✓ | ✓ | $\mathcal{O}(\epsilon^{-2}P^{-1})$ | $\mathcal{O}(\epsilon^{-1})$ |
| ShroFBO (this paper) | ✓ | ✓ | ✓ | ✓ | $\mathcal{O}(\epsilon^{-2}P^{-1})$ | $\mathcal{O}(\epsilon^{-1})$ |

Table 1: Comparison of different federated bilevel algorithms in the setting with heterogeneous data. We do not include the methods with momentum-based acceleration for a fair comparison. $P \leq n$ is the number of sampled clients in each communication round.

**Convergence analysis and improved complexity.** As shown in Table 1, our SimFBO and ShroFBO both achieve a sample complexity (i.e., the number of samples needed to reach an $\epsilon$-accurate stationary point) of $\mathcal{O}(\epsilon^{-2}P^{-1})$, which matches the best result obtained by FedMBO [24] but under a more practical client sampling without replacement. Moreover, SimFBO and ShroFBO both achieve the best communication complexity (i.e., the number of communication rounds to reach an $\epsilon$-accurate stationary point) of $\mathcal{O}(\epsilon^{-1})$, which improves those of other methods by an order of $\epsilon^{-1/2}$. Technically, we develop novel analysis in characterizing the client drifts by the three variables, bounding the per-iteration progress in the global $y$ and $v$ updates, and proving the smoothness and bounded variance in local $v$ updates via induction, which may be of independent interest.

**Superior performance in practice.** In the experiments, the proposed SimFBO method significantly improves over existing strong federated bilevel baselines such as AggITD, FedNest and LFedNest in both the i.i.d. and non-i.i.d. settings. We also validate the better performance of ShroFBO in the presence of heterogeneous local computation due to the resilient server-side updates.

## 2  SimFBO: A Simple and Flexible Framework

### 2.1  Preliminary: Federated Hypergradient Computation

The biggest challenge in FBO is to compute the federated hypergradient $\nabla\Phi(x)$ due to the implicit and complex dependence of $y^*(x)$ on $x$. Under suitable assumptions and using the implicit function theorem in [18], it has been shown that the $\nabla\Phi(x)$ takes the form of

$$\nabla\Phi(x) = \sum_{i=1}^{n} p_i \nabla_x f_i(x, y^*) - \nabla_{xy}^2 G(x, y^*)\left[\nabla_{yy}^2 G(x, y^*)\right]^{-1} \sum_{i=1}^{n} p_i \nabla_y f_i(x, y^*)$$

which poses several computational challenges in the federated setting. First, the second term at the right-hand side contains three global components in a nonlinear manner, and hence the direct aggregation of local hypergradients given by

$$\sum_{i=1}^{n} p_i\left(\nabla_x f_i(x, y^*) - \nabla_{xy}^2 g_i(x, y^*)\left[\nabla_{yy}^2 g_i(x, y^*)\right]^{-1} \nabla_y f_i(x, y^*)\right)$$

is a biased estimation of $\nabla\Phi(x)$ due to the client drift. Second, it is infeasible to compute, store and communicate the second-order Hessian-inverse and Jacobian matrices due to the limited computing and communication resource. Although various AID- and ITD-based approaches have been proposed to address these challenges, they still suffer from several limitations (as we point out in the introduction) such as complicated implementation, high communication cost, lack of client sampling without replacement, and vulnerability to the system-level heterogeneity. To this end, we propose a simple, flexible and communication-efficient FBO framework named SimFBO in this section.

### 2.2  Federated Hypergradient Surrogate

To estimate the federated hypergradient efficiently, we use the surrogate $\bar{\nabla}F(x, y, v) = \nabla_x F(x, y) - \nabla_{xy}^2 G(x, y)v$, where $v \in R^{d_y}$ is an auxiliary vector. Then, it suffices to find $y$ and $v$ as efficient

estimates of the solutions to the global lower-level problem and the global linear system (LS) $\nabla^2_{yy} G(x, y)v = \nabla_y F(x, y)$ that is equivalent to solving following quadratic programming.

$$
\min_v \; R(x, y, v) = \frac{1}{2} v^T \nabla^2_{yy} G(x, y)v - v^T \nabla_y F(x, y)
$$

$$
= \sum_{i=1}^n p_i \big( \underbrace{\frac{1}{2} v^T \nabla^2_{yy} g_i(x, y)v - v^T \nabla_y f_i(x, y)}_{R_i(x,y,v)} \big), \tag{2}
$$

where $R_i(x, y, v)$ can be regarded as the loss function of client $i$ for solving this global LS problem. Based on this surrogate, we next describe the proposed SimFBO framework.

### 2.3 Simple Local and Server-side Aggregations and Updates

**Simple local update.** Differently from FedNest [65] and AggITD [69] that perform the lower-level optimization, the federated hypergradient estimation and the upper-level update alternatively in different communication sub-loops, our SimFBO conducts the simple updates on all these three procedures simultaneously in each communication round. In specific, each communication round $t$ first selects a subset $C^{(t)}$ of participating clients without replacement. Then, each active client $i \in C^{(t)}$ updates three variables $y, v, x$ at $k^{th}$ local iteration **simultaneously** as

$$
\begin{pmatrix} y_i^{(t,k+1)} \\ v_i^{(t,k+1)} \\ x_i^{(t,k+1)} \end{pmatrix} \leftarrow \begin{pmatrix} y_i^{(t,k)} \\ v_i^{(t,k)} \\ x_i^{(t,k)} \end{pmatrix} - a_i^{(t,k)} \begin{pmatrix} \eta_y \nabla_y g_i\big(x_i^{(t,k)}, y_i^{(t,k)}; \zeta_i^{(t,k)}\big) \\ \eta_v \nabla_v R_i\big(x_i^{(t,k)}, y_i^{(t,k)}, v_i^{(t,k)}; \psi_i^{(t,k)}\big) \\ \eta_x \bar{\nabla} f_i\big(x_i^{(t,k)}, y_i^{(t,k)}, v_i^{(t,k)}; \xi_i^{(t,k)}\big) \end{pmatrix} \tag{3}
$$

where $\eta_y, \eta_v, \eta_x$ correspond to the local stepsizes, $a_i^{(t,k)}$ is a client-specific coefficient to increase the flexibility of the framework, $\zeta_i^{(t,k)}, \psi_i^{(t,k)}, \xi_i^{(t,k)}$ are independent samples, and the local hypergradient estimate takes the form of $\bar{\nabla} f_i(x, y, v; \xi) = \nabla_x f_i(x, y; \xi) - \nabla^2_{xy} g_i(x, y; \xi)v_i$. The variables $y, v$ and $x$ in eq. (3), which optimize the lower-level problem, the LS problem and the upper-level problem, are updated with totally $\tau_i^{(t)}$ local steps. Note that the updates in eq. (3) also allow for parallel computation on $x, v$ and $y$ locally.

**Local and server-side aggregation.** After completing all local updates, the next step is to aggregate such local information on both the client and server sides. As shown in eq. (4), each participating client $i \in C^{(t)}$ aggregates all the local gradients, and then communicate the aggregations $q_{y,i}^{(t)}, q_{v,i}^{(t)}$ and $q_{x,i}^{(t)}$ to the server. Then, on the server side, such local information is further aggregated to be $q_y^{(t)}, q_v^{(t)}$ and $q_x^{(t)}$, which will be used for a subsequent generalized server-side update.

$$
q_y^{(t)} = \sum_{i \in C^{(t)}} \widetilde{p}_i q_{y,i}^{(t)} = \sum_{i \in C^{(t)}} \widetilde{p}_i \sum_{k=0}^{\tau_i - 1} a_i^{(t,k)} \nabla_y g_i\big(x_i^{(t,k)}, y_i^{(t,k)}; \zeta_i^{(t,k)}\big),
$$

$$
q_v^{(t)} = \sum_{i \in C^{(t)}} \widetilde{p}_i q_{v,i}^{(t)} = \sum_{i \in C^{(t)}} \widetilde{p}_i \sum_{k=0}^{\tau_i - 1} a_i^{(t,k)} \nabla_v R_i\big(x_i^{(t,k)}, y_i^{(t,k)} v_i^{(t,k)}; \psi_i^{(t,k)}\big),
$$

$$
\underbrace{q_x^{(t)} = \sum_{i \in C^{(t)}} \widetilde{p}_i q_{x,i}^{(t)}}_{\text{Server aggregation}} = \sum_{i \in C^{(t)}} \widetilde{p}_i \underbrace{\sum_{k=0}^{\tau_i - 1} a_i^{(t,k)} \bar{\nabla} f_i\big(x_i^{(t,k)}, y_i^{(t,k)}, v_i^{(t,k)}; \xi_i^{(t,k)}\big)}_{\text{Local aggregation}}, \tag{4}
$$

where $\widetilde{p}_i := \frac{n}{|C^{(t)}|} p_i$ is the effective weight of client $i \in C^{(t)}$ among all participating clients such that $\mathbb{E}(\sum_{i \in C^{(t)}} \widetilde{p}_i) = 1$. Note that in eq. (4), the local aggregation $q_{y,i}^{(t)}$ (similarly for $v$ and $x$) can be regarded as a linear combination of all local stochastic gradients, and hence covers a variety of local optimizers such as stochastic gradient descent, momentum-based gradient, variance reduction by choosing different coefficients $a_i^{(t,k)}$ for $i \in C^{(t)}$. This substantially enhances the flexibility of the proposed framework.

**Algorithm 1** SimFBO and ShroFBO

---

1: **Input:** initialization $y^{(0)}, v^{(0)}, x^{(0)}$, number of communication rounds $T$, learning rates: client $\{\eta_y, \eta_v, \eta_x\}$, server: $\{\gamma_y, \gamma_v, \gamma_x\}$, local update rounds: $\{\tau_i^{(t)}\}$
2: **for** $t = 0, 1, 2, ..., T$ **do**
3:    **for** $i \in C^{(t)}$ in parallel **do**
4:       $y_i^{(t,0)} = y^{(t)}, v_i^{(t,0)} = v^{(t)}, x_i^{(t,0)} = x^{(t)}$
5:       **for** $k = 0, 1, 2, ..., \tau_i^{(t)} - 1$ **do**
6:          Locally update $y_i^{(t,k)}, v_i^{(t,k)}$ and $x_i^{(t,k)}$ simultaneously via eq. (3)
7:       **end for**
8:       Client $i$ locally aggregates gradients to compute $q_{y,i}^{(t)}, q_{v,i}^{(t)}, q_{x,i}^{(t)}$ via eq. (4)
9:       Client $i$ locally aggregates gradients to compute $h_{y,i}^{(t)}, h_{v,i}^{(t)}, h_{x,i}^{(t)}$ defined in eq. (6)
10:    **end for**
11:    Client $i \in C^{(t)}$ communicate $\{q_{y,i}^{(t)}, q_{v,i}^{(t)}, q_{x,i}^{(t)}\}$ or $\{h_{y,i}^{(t)}, h_{v,i}^{(t)}, h_{x,i}^{(t)}\}$ to the server
12:    Server aggregates local estimators to compute $\{q_y^{(t)}, q_v^{(t)}, q_x^{(t)}\}$ using eq. (4)
13:    Server aggregates local estimators to compute $\{h_y^{(t)}, h_v^{(t)}, h_x^{(t)}\}$ using eq. (7)
14:    Server updates using eq. (5)
15:    Server updates using eq. (8)
16: **end for**

---

**Server-side updates.** Based on the aggregated gradients $q_y^{(t)}, q_v^{(t)}$ and $q_x^{(t)}$, we then perform server-level gradient-based updates on variables $x, v$ and $y$ simultaneously as

$$y^{(t+1)} = y^{(t)} - \gamma_y q_y^{(t)}, \quad v^{(t+1)} = \mathcal{P}_r\big(v^{(t)} - \gamma_v q_v^{(t)}\big), \quad x^{(t+1)} = x^{(t)} - \gamma_x q_x^{(t)}, \qquad (5)$$

where $\gamma_y, \gamma_v$ and $\gamma_x$ are server-side updating stepsizes for $y, v, x$ and $\mathcal{P}_r(v) := \min\big\{1, \frac{r}{\|v\|}\big\}v$ is a simple projection on a bounded ball with a radius of $r$. There are a few remarks about the updates in eq. (5). First, in contrast to existing FBO algorithms such as [65, 69], our introduced server-side updates leverage not only the client-side stepsizes $\eta_y, \eta_v, \eta_x$, but also the server-side stepsizes $\gamma_y, \gamma_v$ and $\gamma_x$. This generalized two-learning-rate paradigm can provide more algorithmic and theoretical flexibility, and provides improved communication efficiency in practice and in theory. Second, the projection $\mathcal{P}_r(\cdot)$ serves as an important step to ensure the boundedness of variable $v^{(t)}$, and hence guarantee the smoothness of the global LS problem and the boundedness of the estimation variance in $v$ and $x$ updates, both of which are crucial and necessary in the final convergence analysis. Note that we do not impose such projection on the local $v_i^{(t,k)}$ variables because we can prove via induction that they are bounded given the boundedness of $v^{(t)}$ (see Proposition 1).

## 2.4 Resilient Server-side Updates against System-level Heterogeneity

**Limitations under system-level heterogeneity.** When clients have heterogeneous computing and storing capabilities (e.g., computer server v.s. phone in edge computing), an unequal number of local updates are often performed such that the global solution can be biased toward those of the clients with much more local steps or stronger optimizers. As observed in [67], this heterogeneity can deviate the iterates to minimize a different objective function. To explain this mismatch phenomenon, inspired by [57], we rewrite the server-side update on $x$ (similarly for $v$ and $y$) in eq. (4) as

$$q_x^{(t)} = \sum_{i=1}^{n} p_i q_i^{(t,k)} = \underbrace{\Big(\sum_{j=1}^{n} p_j \|a_j^{(t)}\|_1\Big)}_{\rho^{(t)}} \sum_{i=1}^{n} \underbrace{\frac{p_i \|a_i^{(t)}\|_1}{\sum_{j=1}^{n} p_j \|a_j^{(t)}\|_1}}_{w_i} \underbrace{\frac{q_{x,i}^{(t)}}{\|a_i^{(t)}\|_1}}_{h_{x,i}^{(t)}}. \qquad (6)$$

where $a_i^{(t)} = \big[a_i^{(t,0)}, ..., a_i^{(t,\tau_i^{(t)}-1)}\big]^T$ collects all local coefficients of client $i$, and $h_{x,i}^{(t)}$ **normalizes** the aggregated gradient $q_{x,i}^{(t)}$ by $1/\|a_i^{(t)}\|_1$ such that $\|h_{x,i}^{(t)}\|$ does not grow with the increasing of $\tau_i^{(t)}$.

Although such normalization can help to mitigate the system-level heterogeneity, the effective weight $w_i$ can deviate from the true weight $p_i$ of the original objective in eq. (1), and the iterates converge to the stationary point of a different objective that replaces all $p_i$ by $w_i$ in eq. (1) (see Theorem 1).

**System-level heterogeneity robust FBO (ShroFBO).** To address this convergence issue, we then propose a new method named ShroFBO with stronger resilience to such heterogeneity. Motivated by the normalized reformulation in eq. (6), ShroFBO adopts a different server-side aggregation as

$$h_y^{(t)} = \sum_{i \in C^{(t)}} \widetilde{p}_i h_{y,i}^{(t)}, \quad h_v^{(t)} = \sum_{i \in C^{(t)}} \widetilde{p}_i h_{v,i}^{(t)}, \quad h_x^{(t)} = \sum_{i \in C^{(t)}} \widetilde{p}_i h_{x,i}^{(t)}, \tag{7}$$

where $\widetilde{p}_i := \frac{n}{|C^{(t)}|} p_i$ and $h_{y,i}^{(t)}, h_{v,i}^{(t)}, h_{x,i}^{(t)}$ are the normalized local aggregations defined in eq. (6). Accordingly, the server-side updates become

$$y^{(t+1)} = y^{(t)} - \rho^{(t)} \gamma_y h_y^{(t)}, \quad v^{(t+1)} = \mathcal{P}_r\big(v^{(t)} - \rho^{(t)} \gamma_v h_v^{(t)}\big), \quad x^{(t+1)} = x^{(t)} - \rho^{(t)} \gamma_x h_x^{(t)}. \tag{8}$$

Differently from eq. (6), we select the client weights to be $\widetilde{p}_i$ to enforce the correct convergence to the stationary point of the original objective in eq. (1), as shown in Theorem 2 later.

## 3 Main Result

### 3.1 Assumptions and Definitions

We make the following standard definitions and assumptions for the outer- and inner-level objective functions, as also adopted in stochastic bilevel optimization [26, 21, 30] and in federated bilevel optimization [65, 69, 24].

**Definition 1.** *A mapping $F$ is $L$-Lipschitz continuous if for $\forall z, z'$, $\|F(z) - F(z')\| \le L\|z - z'\|$.*

Since the overall objective $\Phi(x)$ is nonconvex, the goal is expected to find an $\epsilon$-accurate stationary point defined as follows.

**Definition 2.** *We say $z$ is an $\epsilon$-accurate stationary point of the objective function $\Phi(x)$ if $\mathbb{E}\|\nabla\Phi(z)\|^2 \le \epsilon$, where $z$ is the output of an algorithm.*

**Assumption 1.** *For any $x \in \mathbb{R}^{d_x}$, $y \in \mathbb{R}^{d_y}$ and $i \in \{1, 2, ..., n\}$, $f_i(x,y)$ and $g_i(x,y)$ are twice continuously differentiable, and $g_i(x,y)$ is $\mu_g$-strongly convex w.r.t. $y$.*

The following assumption imposes the Lipschitz continuity conditions on the upper- and lower-level objective functions and their derivatives.

**Assumption 2.** *Function $f_i(x,y)$ is $L_f$-Lipschitz continuous; the gradients $\nabla f_i(x,y)$ and $\nabla g_i(x,y)$ are $L_1$-Lipschitz continuous; the second-order derivatives $\nabla^2 f_i(x,y)$ and $\nabla^2 g_i(x,y)$ are $L_2$-Lipschitz continuous; and the third-order derivatives $\nabla^3 g_i(x,y)$ is $L_3$-Lipschitz continuous for some constants $L_f, L_1, L_2, L_3 > 0$.*

The Lipschitz continuity of the third-order derivative is necessary here to ensure the smoothness of $v^*(x)$, which guarantees the descent in the iterations of LS function (see Lemma 10), under our more challenging simultaneous and single-loop updating structure. Next, we assume the bounded variance conditions on the gradients and second-order derivatives.

**Assumption 3.** *There exist constants $\sigma_f^2, \sigma_g^2, \sigma_{gg}^2$ such that $\mathbb{E}\big[\|\nabla f_i(x,y) - \nabla f_i(x,y;\xi)\|^2\big] \le \sigma_f^2$, $\mathbb{E}\big[\|\nabla g_i(x,y) - \nabla g_i(x,y;\zeta)\|^2\big] \le \sigma_g^2$ and $\mathbb{E}\big[\|\nabla^2 g_i(x,y) - \nabla^2 g_i(x,y;\zeta)\|^2\big] \le \sigma_{gg}^2$.*

**Assumption 4.** *For any $x \in \mathbb{R}^{d_x}$, $y \in \mathbb{R}^{d_y}$, there exist constants $\beta_{gh} \ge 1$ and $\sigma_{gh} \ge 0$ such that*

$$\sum_{i=1}^n w_i \|\nabla_y g_i(x,y)\|^2 \le \beta_{gh}^2 \|\sum_{i=1}^n w_i \nabla_y g_i(x,y)\|^2 + \sigma_{gh}^2.$$

*We have $\beta_{gh} = 1$, and $\sigma_{gh} = 0$ when all $g_i$'s are identical.*

This assumption of global heterogeneity uses $\beta_{gh}$ and $\sigma_{gh}$ to measure the dissimilarity of $\nabla_y g_i(x,y)$ for all $i$.

## 3.2 Convergence and Complexity Analysis

It can be seen from eq. (2) that the boundedness of $v$ is necessary to guarantee the smoothness (w.r.t. $x, y$) and bounded variance in solving the local and global LS problems. Projecting the global $v^{(t)}$ vector and the local $v_i^{(t,k)}, k \geq 1$ vectors onto a bounded set can be a feasible solution, but in this case, the local aggregation $q_{v,i}^{(t)}$ is no longer a linear combination of local gradients. This can complicate the implementation and analysis, and degrade the flexibility of the framework. Fortunately, we show via induction that the projection of the server-side vector $v^{(t)}$ on a bounded set suffices to guarantee the boundedness of local vectors $v_i^{t,k}$.

**Proposition 1** (Boundedness of Local $v$). *Under Assumptions 1 and 2, for each iteration $t$, client $i$, and local iteration $k = 1, 2, ..., \tau_i^{(t)}$, we have $r_i := \|v_i^{(t,k)}\| \leq \left(1 + \frac{\alpha_{\max}}{\alpha_{\min}}\right) r$, where the radius $r = \frac{L_f}{\mu_g}$ and $\alpha_{\min}, \alpha_{\max}$ are chosen such that $\alpha_{\min} \leq a_i^{(t,k)} \leq \alpha_{\max}$.*

Next, we show an important proposition in characterizing the per-iteration progress of the global $v^{(t)}$ updates in approximating the solution of a reweighted global LS problem. Let $\Delta_v^{(t)} = \mathbb{E}\|v^{(t)} - \widetilde{v}^*(x^{(t)})\|^2$ denote the approximation error, where $\widetilde{v}^*$ be the minimizer of $\sum_{i=1}^n w_i R_i(x, \widetilde{y}^*, \cdot)$.

**Proposition 2.** *Under the Assumption 1, 2 and 3, the iterates $v^{(t)}$ in solving the global LS problem generated by Algorithm 1 satisfy*

$$\mathbb{E}\|v^{(t+1)} - \widetilde{v}^*(x^{(t+1)})\|^2 - \mathbb{E}\|v^{(t)} - \widetilde{v}^*(x^{(t)})\|^2$$

$$\leq (\delta_t' - \rho^{(t)}\gamma_v\mu_g - \delta_t'\rho^{(t)}\gamma_v\mu_g)\mathbb{E}\|v^{(t)} - \widetilde{v}^*(x^{(t)})\|^2 + (1 + \delta_t')(\rho^{(t)}\gamma_v)^2\mathbb{E}\left\|\sum_{i \in C^{(t)}} \widetilde{w}_i h_{v,i}^{(t)}\right\|^2$$

$$+ (1 + \delta_t')\rho^{(t)}\gamma_v \frac{4L_R^2}{\mu_g}\mathbb{E}\|y^{(t)} - \widetilde{y}^*(x^{(t)})\|^2 + (\rho^{(t)}\gamma_x)^2\left(L_v^2 + \frac{L_{vx}}{4}\right)\mathbb{E}\left\|\sum_{i \in C^{(t)}} \widetilde{w}_i h_{x,i}^{(t)}\right\|^2$$

$$+ (1 + \delta_t')\rho^{(t)}\gamma_v \frac{4L_R^2}{\mu_g} \sum_{i=1}^n w_i \sum_{k=0}^{\tau_i-1} \frac{a_i^{(t,k)}}{\|a_i^{(t)}\|_1}\mathbb{E}\left[\|x^{(t)} - x_i^{(t,k)}\|^2 + \|y^{(t)} - y_i^{(t,k)}\|^2\right.$$

$$+ \left.\|v^{(t)} - v_i^{(t,k)}\|^2\right] + (\rho^{(t)}\gamma_x)^2 \frac{2L_v}{\delta_{t,1}'}\mathbb{E}\left\|\sum_{i=1}^n w_i\widetilde{h}_{x,i}^{(t)}\right\|^2.$$

*for all $t \in \{0, 1, ..., T-1\}$, $k \in \{0, 1, ..., \tau_i^{(t)} - 1\}$ and $i \in \{1, 2, ..., n\}$, where $\widetilde{w}_i := \frac{n}{|C^{(t)}|}w_i$.*

Similarly, we can provide a per-iteration process of $y^{(t)}$ in approximating the solution $\widetilde{y}^*$ of the reweighted lower-level global function $\sum_{i=1}^n w_i g_i(x, \cdot)$. Note that such characterizations do not exist in previous studies in single-level or minimax federated optimization with a single objective (e.g., [57]) because our analysis needs to handle three different lower-level, LS and upper-level objectives. As shown in Proposition 2, the bound involves the client drift term $\mathbb{E}\|v^{(t)} - v_i^{(t,k)}\|^2$ (similarly for $y, x$), so the next step is to characterize this important quantity.

**Proposition 3.** *Under Assumption 1 and 2, the local iterates client drift of $v_i^{(t,k)}$ is bounded as*

$$\sum_{i=1}^n w_i \frac{1}{\|a_i^{(t)}\|_1} \sum_{k=1}^{\tau_i-1} a_i^{(t,k)}\mathbb{E}\|v_i^{(t,k)} - v^{(t)}\|^2 \leq \eta_v^2\bar{\tau}\sigma_{M1}^2,$$

*for all $t \in \{0, 1, ..., T-1\}$, $k \in \{0, 1, ..., \tau_i - 1\}$ and $i \in \{1, 2, ..., n\}$. We define $\bar{\tau} := \sum_{i=1}^n \tau_i/n$ and $\sigma_{M1}^2 := \alpha_{\max}^2(\sigma_f^2 + r_{\max}^2\sigma_{gg}^2) + \alpha_{\max}(L_f^2 + r_{\max}^2L_1^2)$.*

It can be seen from Proposition 3 that the bound on the client drift of the local updates on $v$ is proportional to $\eta_v$ and $\|a_i^{(t)}\|_1$. Since $\alpha_{\min} \leq a_i^{(t,k)} \leq \alpha_{\max}$, $\|a_i^{(t)}\|_1$ is proportional to the number $\tau_i^{(t)}$ of local steps. Thus, this client drift is controllable by choosing $\tau_i^{(t)}$ and the local stepsizes $\eta_v$ properly. Then, combining the results in the above Proposition 1, 2, 3, and under a proper Lyapunov function, we obtain the following theorem. Let $P = |C^{(t)}|$ be the number of sampled clients.

**Theorem 1.** *Define $\widetilde{\Phi}(x) = \widetilde{F}(x, \widetilde{y}^*)$ as the objective function by replacing $p_i$ in eq. (1) with $w_i$. Suppose Assumptions 1, 2 and 3 are satisfied. The iterates by SimFBO in Algorithm 1 satisfy*

$$\min_t \mathbb{E}\left\|\nabla\widetilde{\Phi}(x^{(t)})\right\|^2 = \underbrace{\mathcal{O}\left(\frac{M_1(n-P)}{n}\sqrt{\frac{\bar{\tau}}{PT}}\right)}_{\text{partial participation error}} + \underbrace{\mathcal{O}\left(M_2\sqrt{\frac{1}{P\bar{\tau}T}}\right)}_{\text{full synchronization error}} + \underbrace{\mathcal{O}\left(\frac{M_3}{\bar{\tau}T}\right)}_{\text{local updates error}}, \quad (9)$$

*where $\gamma_x, \gamma_y, \gamma_v, \eta_x, \eta_y, \eta_v$ are set in eq. (38) and $M_1, M_2, M_3$ are defined by eq. (40) in appendix. For the full client participation (i.e., $P = n$), the sample complexity is $\bar{\tau}T = \mathcal{O}(n^{-1}\epsilon^{-2})$, and the number of communication rounds is $T = \mathcal{O}(\epsilon^{-1})$. For partial client participation, the sample complexity is $\bar{\tau}T = \mathcal{O}(P^{-1}\epsilon^{-2})$, and the number of communication rounds is $T = \mathcal{O}(P^{-1}\epsilon^{-2})$.*

First, when set $\bar{\tau} = \mathcal{O}(1)$, Theorem 1 shows that SimFBO converges to a stationary point of an objective function $\widetilde{\Phi}(x)$ with a rate of $\mathcal{O}(\frac{1}{\sqrt{PT}} + \frac{1}{T})$, which, to the best of our knowledge, is the first linear speedup result under partial client participation without replacement. Note that without system-level heterogeneity, i.e., $\|a_1^{(t)}\| = ... = \|a_n^{(t)}\|$, $w_i = \frac{p_i\|a_i^{(t)}\|_1}{\sum_{j=1}^n p_j\|a_j^{(t)}\|_1} = p_i$, and hence SimFBO converges to the stationary point of the original objective in eq. (1). However, in the presence of system-level heterogeneity, SimFBO may converge to the stationary point of a different objective. Second, when nearly full clients participate, the partial participation error is approximately zero. Then we can see that setting local update round $\bar{\tau}$ to its upper-bound results in the best performance.

**Theorem 2.** *Define $\Phi(x) = F(x, y^*)$ as eq. (1). Suppose Assumptions 1, 2 and 3 are satisfied. The iterates generated by ShroFBO in Algorithm 1 satisfy*

$$\min_t \mathbb{E}\left\|\nabla\Phi(x^{(t)})\right\|^2 = \mathcal{O}\left(\frac{M_1(n-P)}{n}\sqrt{\frac{\bar{\tau}}{PT}}\right) + \mathcal{O}\left(M_2\sqrt{\frac{1}{P\bar{\tau}T}}\right) + \mathcal{O}\left(\frac{M_3}{\bar{\tau}T}\right), \quad (10)$$

*by setting the same server-side and local stepsizes and $M_1, M_2$ and $M_3$ as in Theorem 1. For full client participation, the sample complexity is $\bar{\tau}T = \mathcal{O}(n^{-1}\epsilon^{-2})$, and the number of communication rounds is $T = \mathcal{O}(\epsilon^{-1})$. For partial client participation, the sample complexity is $\bar{\tau}T = \mathcal{O}(P^{-1}\epsilon^{-2})$, and the number of communication rounds is $T = \mathcal{O}(P^{-1}\epsilon^{-2})$.*

In Theorem 2, we show that even under the system-level heterogeneity, ShroFBO can converge to the original objective function with the same convergence rate as SimFBO. This justifies the design principle of robust server-side updates.

## 4 Related Work

**Bilevel optimization.** Bilevel optimization, first introduced by [3], has been studied for decades. A class of constraint-based bilevel methods was then proposed [20, 16, 59, 61], whose idea is to replace the lower-level problem by the optimality conditions. Gradient-based bilevel algorithms have attracted considerable attention due to the effectiveness in machine learning. Among them, AID-based approaches [8, 51, 38, 1] leveraged the implicit derivation of the hypergradient, which was then approximated via solving a linear system. ITD-based approaches [45, 12, 11, 56, 17] approximated the hypergradient based on automatic differentiation via the forward or backward mode. A group of stochastic bilevel approaches has been developed and analyzed recently based on Neumann series [5, 26, 1], recursive momentum [72, 23, 19] and variance reduction [72, 7], etc. For the lower-level problem with multiple solutions, several approaches were proposed based on the upper- and lower-level gradient aggregation [55, 43, 34], barrier types of regularization [41, 39], penalty-based formulations [58], primal-dual technique [62], and dynamic system-based methods [42].

**Federated (bilevel) learning.** Federated Learning was proposed to enable collaborative model training across multiple clients without compromising the confidentiality of individual data [32, 60, 49]. As one of the earliest methods of federated learning [47], FedAvg has inspired an increasing number of approaches to deal with different limitations such as slower convergence, high communication cost and undesired client drift by leveraging the techniques including proximal regularization [37], periodic variance reduction [48, 28], proximal splitting [50], adaptive gradients [53]. Theoretically, the convergence of FedAvg and its variants has been analyzed in various settings with the homogeneous [63, 68, 64, 2] or heterogeneous datasets [37, 66, 48, 29]. [67] analyzed the impact of the

system-level heterogeneity such as heterogeneous local computing on the convergence. [57] further extended the analysis and the methods to the minimax problem setting.

Federated bilevel optimization has not been explored well except for a few attempts recently. For example, [14, 35] proposed momentum-based bilevel algorithms, and analyzed their convergence in the setting with homogeneous datasets. In the setting with non-i.i.d. datasets, [65] and [24] proposed FedNest and FedMBO via AID-based federated hypergradient estimation, and [69] proposed an ITD-based aggregated approach named Agg-ITD. Momentum-based techniques have been also used by [22, 36] to improve the sample complexity. Moreover, there are some studies that focus on other distributed scenarios, including decentralized bilevel optimization [6, 73, 44], asynchronous bilevel optimization over directed network [74], and distributed bilevel network utility maximization [27].

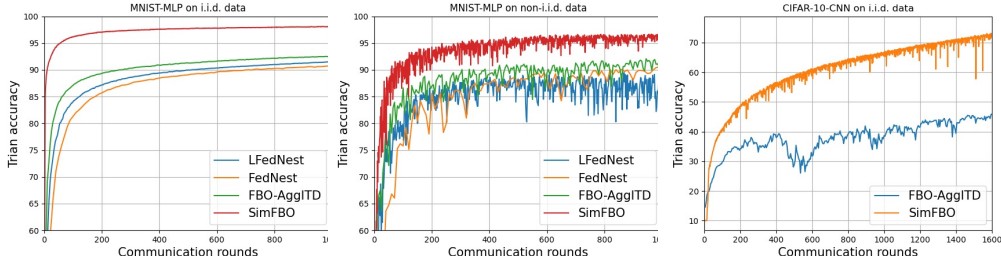

Figure 2: Comparison among our SimFBO, FBO-AggITD [69], FedNest [65] and LFedNest [65]. The left and middle ones plot the training accuracy v.s. # of communication rounds on i.i.d. MNIST datasets with MLP networks, and the right one plots the training accuracy v.s. # of rounds on i.i.d. CIFAR-10 datasets with a 7-layer CNN.

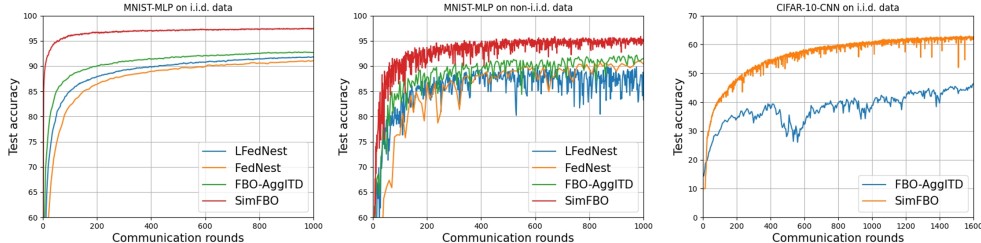

Figure 3: Comparison of different methods: the test accuracy v.s. # of communication rounds.

## 5  Experiments

In this section, we perform two hyper-representation experiments to compare the performance of our proposed SimFBO algorithm with FBO-AggITD [69], FedNest [69], and LFedNest [65], and validate the better performance of ShroFBO in the presence of heterogeneous local computation. We test the performance on MNIST and CIFAR datasets with MLP and CNN backbones. We follow the same experimental setup and problem formulation as in [65, 69]. The details of all experimental specifications can be found in Appendix A.1.

**Comparison to existing methods.** The comparison results are presented in Figure 2 and Figure 3. It can be seen that across different datasets and backbones, our proposed SimFBO consistently converges much faster than other comparison methods, while achieving a much higher training and test accuracy. We do not plot the curves of FedNest and LFedNest on CIFAR and CNN, because they are hard to converge under various hyperparameter configurations using their source codes.

**Performance under heterogeneous local computation.** We now test the performance in the setting where a total of 10 clients perform a variable number of local steps. This is to simulate the scenario where clients have heterogeneous computing capabilities and hence can perform an uneven number of local updates. In this experiment, we choose the number $\tau_i$ of the client $i$'s local steps from the set $\{1, ..., 10\}$ uniform at random. As shown in Figure 4, the proposed ShroFBO method performs the best due to the better resilience to such client heterogeneity. We also compare the convergence rate of our proposed SimFBO, FedNest and FBO-AggITD w.r.t. running time. The results are provided in Figure 5. All the settings for different algorithms are the same as in Appendix A.2. It can be seen that the proposed SimFBO still converges fastest with a higher test accuracy in terms of running time.

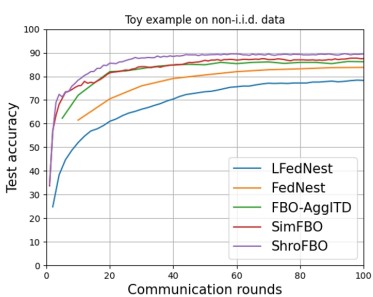 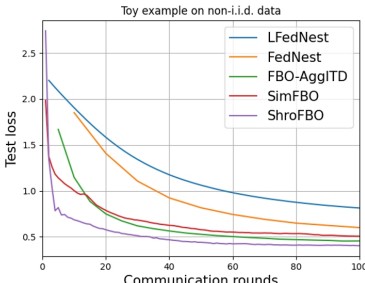

Figure 4: Comparison among different algorithms in the presence of heterogeneous local computation.

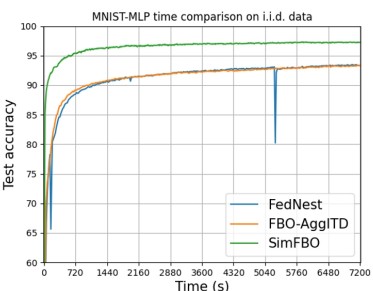 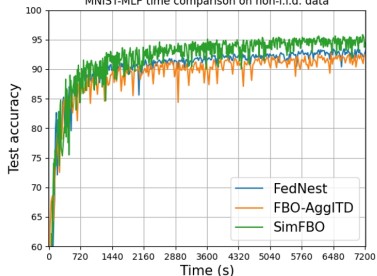

Figure 5: Time Comparison among different algorithms.

## 6 Conclusion

In this paper, we propose a simple and communication-efficient federated bilevel algorithm named SimFBO and its variant ShroFBO with better resilience to the system-level heterogeneity. We show that both SimFBO and ShroFBO allow for more practical client sampling without replacement, and achieve better sample and communication complexities. Experiments demonstrate the great promise of the proposed methods. We anticipate that the proposed algorithms and the developed analysis can be applied to other distributed settings such as decentralized or asynchronous bilevel optimization, and the algorithms may be useful in applications such as hyperparameter tuning or fine-tuning in federated learning or AI-aware edge computing.

## Acknowledgement

The work is supported in part by NSF under grants 2326592 and 2311274.

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
