# OpenReview forum: "SimFBO: Towards Simple, Flexible and Communication-efficient Federated Bilevel Learning"
_NeurIPS.cc/2023/Conference — NeurIPS 2023 spotlight_

### Official Review · Reviewer_DZHN · 2023-07-06

**Soundness:** 3 good
**Presentation:** 4 excellent
**Contribution:** 3 good
**Rating:** 7
**Confidence:** 3

**Summary:**

This paper focuses on bilevel optimization in a federated learning environment. Bilevel optimization has various applications in federated learning (FL) and few recent works proposed versions of bilevel optimization schemes for FL. A challenging step in bilevel optimization is the computation of the "hypergradient", and the existing schemes are able to obtain an estimate of the hypergradient, albeit a biased estimate, with a substantial communication overhead (multiple rounds of communication per server-side update). The paper reformulates the hypergradient computation as a least squares problem that can provide an unbiased estimate of the hypergradient, while requiring only a single round of communication for every server-side update. Based on this estimate, the paper presents a federated bilevel optimization algorithm, SimFBO, and a variant robust to system-level heterogeneity, ShroFBO. The theoretical analyses demonstrate that the proposed algorithms can converge with a sample complexity comparable to that of existing federated bilevel schemes, while demonstrating a significant improvement in the communication overhead. Preliminary empirical results highlight the significant communication efficiency of the proposed schemes compared to the existing baselines.

**Strengths:**

**Well-motivated intuitive presentation.**
The authors do an excellent job in presenting this paper. The main problem and the challenges with the existing solutions are clearly discussed, and it is easy to see why the existing schemes are not very practical. The main idea of the use of the global least-squares reformulation, while obvious in hindsight, is very well presented and motivated, making it easy for the reader to follow along and realize how the existing challenges are mitigated. The theoretical results are clearly and intuitively presented with proper discussion highlighting the main parts of the analysis, and the main steps (with appropriate pointers to the supplement).


**Critical reformulation removing both bias and communication overhead.**
A key strength of this paper is a simple (yet of significant practical impact) reformulation of the hypergradient estimation using a standard quadratic program. A key property that the authors leverage is the fact that the global quadratic objective can be decomposed into per-client quadratic objectives, which is not true of the global hypergradient (which cannot be decomposed into per-client hypergradients). This simple yet powerful insight is then utilized to obtain an estimate which, upon proper solution of the global least-squares problem, is unbiased, and can be efficiently updated along side the upper and lower level variables in the bilevel problem. While this global least-squares reformulation does facilitate an intuitive communication-efficient algorithm, the paper also performs a thorough theoretical analysis, demonstrating how the inaccuracy in the hypergradient estimate plays into the convergence. The overall algorithm makes the solution of federated bilevel problems significantly more practical.


**Generality of the proposed algorithmic framework.**
The authors do a great job at highlighting the generality of the algorithm framework. First, the general client and server aggregation (in equation (4)) allows us to cover various different client side optimizers, and the analysis is able to provide a convergence guarantee with such generalized aggregations. Second, the proposed framework incorporates system-level heterogeneity, allowing for different clients to perform different levels of client-side updates, and making the server-side aggregation robust to such heterogeneity. This robustness is demonstrated empirically, and the robust version of the algorithm is analysed theoretically.


**Weaknesses:**

**Increased hyperparameter space.**
The proposed framework utilizes various hyperparameters:
- The client-side learning rates $\eta_{y/v/x}$ and iterations $\tau_i^{(t)}$ (for each server side update $t \in [T]$ and each client $i \in [n]$)
- The server-side learning rates $\gamma_{y/v/x}$
- Potentially the choice of the client-side coefficients $a_i^{(t)} = \left[a_i^{(t,0)}, \ldots, a_i^{\left(t, \tau_i^{(t)}-1 \right)} \right]^\top$ (for each client $i \in [n]$), which might be potentially tied to $\alpha_\min, \alpha_\max$.
- The hypergradient projection radius $r$

As per the theoretical analyses, it can be seen that the best convergence rate of any execution will critically depend on an appropriate setting of these problem-dependent hyperparameters. Since these hyperparameters often depend on quantities that cannot be efficiently estimated (such as Lipschitz constant), the practical bilevel implementations usually utilize some form of hyperparameter search. Hyperparameter optimization is known to be a hard unsolved problem in FL because of the overall communication overhead. This makes it hard to see how the proposed federated bilevel framework can live up to its practical potential -- one can view this proposed federated bilevel framework as having shifted the communication overhead from the model training stage to the hyperparameter optimization stage, without reducing the overall communication necessary for good training convergence (which involves trying various hyperparameters and training with them).

**Questions:**

- In lines 87-88, it seems that $\tilde{v}^*(t)$ in the definition of $\Delta_v^{(t)}$ depends on the iteration $t$. However, line 88 claims that $\tilde{v}^*$ is the minimizer of $\sum_{i=1}^n w_i R_i(x, y^*, \cdot)$, which implies that $\tilde{v}^*(t)$ is not dependent on $t$. Can this be clarified? Usually in the bilevel analysis, we are tracking quantities such as $|| y^*(x^{(t)}) - y^{(t+1)} ||$ for dependent variables, such as $y$ and $v$. So $\tilde{v}^*(t)$ can also be the minimizer of $\sum_{i=1}^n w_i R_i(x^{(t)}, y^*(x^{(t)}), \cdot)$ or even $\sum_{i=1}^n w_i R_i(x^{(t)}, y^{(t+1)}, \cdot)$.

- What is $\bar{\tau}$ in Theorem 1? It seems to be introduced in the theorem statement, but I am not unable to find (in the main paper) what this $\bar{\tau}$ is supposed to signify. Is it some aggregate of the $\{ \tau_i^{(t)} \}$ across all $i \in [n]$ and $t \in [T]$?

Minor:
- The legend in Figure 2 (right) seems off since it has no SimFBO.

**Limitations:**

I did not find any discussion on limitations in the main paper. However, I do not anticipate any potential negative societal impact of this work.

---

> ### Author Rebuttal · Authors · 2023-08-09
>
> We thank the reviewer DZHN for the time and valuable feedback!
>
> **Q1: Hyperparameter optimization is known to be a hard unsolved problem in FL because of the overall communication overhead. This makes it hard to see how the proposed federated bilevel framework can live up to its practical potential -- one can view this proposed federated bilevel framework as having shifted the communication overhead from the model training stage to the hyperparameter optimization stage, without reducing the overall communication necessary for good training convergence (which involves trying various hyperparameters and training with them).**
>
> **A:** Thanks for pointing this out to us! In our practical implementation, we set the client-side learning rates to be the same as $\eta$ for the updates on $y,v,x$ (similarly we set the server-side learning rates to $\gamma$). In addition, we set the iterations $\tau_i^{(t)}$ to be the same as $\tau$ for all clients $i\in [n]$ and all $t\in [T]$. The coefficients $a_i^{(t)}$ are only used to show the flexibility of our framework to cover various local optimizers, and hence we simply set all $a_i^{(t,k)} = 1$ in the experiments. The purpose of radius $r$ is only to guarantee the boundedness of $v$ updates, and hence any sufficiently large constant suffices. In summary, our implementation involves only three critical hyperparameters $\eta$, $\gamma$ and $\tau$, and hence is still quite efficient in practice.
>
>
> **Q2: In line 187-188,** $v^*$ **is defined unclearly.**
>
> **A:** We guess you refer to lines 187-188. Sorry about the confusion. To make this clearer, we revise this sentence as follows:
> Let $\Delta_v^{(t)} := \mathbb{E}\\|v^{(t)} - \widetilde{v}^*(t)\\|^2$,
> $\widetilde{v}^*(t) = \arg\min_v \sum_{i=1}^nw_iR_i(x^{(t)}, y^*(x^{(t)}), v)$ denote the approximation error, where $\widetilde{v}^*(x^{(t)})$ is the minimizer of $\sum_{i=1}^nw_iR_i(x^{(t)}, y^*(x^{(t)}), \cdot)$.
> Then, it can be seen that $\widetilde{v}^*(t)$ is dependent on $t$. Please let us know if we do not understand your question correctly.
>
> **Q3: What is **$\bar{\tau}$** in Theorem 1?  Is it some aggregate of the** $\tau_i^{(t)}$ **across all** $i\in[n]$ **and** $t\in [T]$**?**
>
> **A:** Yes, you are right. In the analysis, we set all $\tau_i^{(t)}$ to be $\tau_i$  independent of time $t$. Then, $\bar{\tau}$ is defined as $\bar{\tau} = \sum_i^n w_i \tau_i$, as given in line 614. We will clarify it in the revision.

---

> > ### Comment · Reviewer_DZHN · 2023-08-21
> >
> > Thank you for the response. My questions regarding the terms are clarified. Regarding the discussion on hyperparameters, it is good to understand that there are only effectively three hyperparameters. I have no further questions.

---

> > > ### Author Response · Authors · 2023-08-21
> > > **Thanks for your feedback**
> > >
> > > Dear reviewer DZHN,
> > >
> > > Thank you so much for your great time and efforts in the review and rebuttal! We really appreciate your suggestions, and will definitely incorporate them in our revision.
> > >
> > > Best,
> > > Authors

---

### Official Review · Reviewer_TKNN · 2023-07-06

**Soundness:** 4 excellent
**Presentation:** 4 excellent
**Contribution:** 4 excellent
**Rating:** 8
**Confidence:** 4

**Summary:**

This work consider the federated bilevel optimization problem. Compared to existing methods, the authors develop a new and simple method named SimFBO without subloops and requiring much fewer communication rounds at each iteration. In the setting with system-level heterogeneity like diverse local steps, they further propose a more robust version named ShroFBO that is shown with a correct convergence under such heterogeneity. Convergence analysis is developed for both methods. Empirical results have shown the great performance of the proposed methods.

**Strengths:**

1. Federated bilevel optimization is a relatively new but challenging problem even when the lower-level objective is strongly convex. This work provides a reasonable and quite effective framework to this problem.

2. The proposed algorithms are novel, easy to understand and theoretically grounded. Providing a simple solution with strong empirical and theoretical performance is strong.

3. Technical derivations seem to be nontrivial. For example, it is interesting to deal with the challenges by client drift, boundness of local iterates etc. It seems to fill the gap of communication complexity of $1/\epsilon$ in federated bilevel problems. This is good.

**Weaknesses:**

The proposed methods require second-order derivatives, which may cause some scalability issue. Is it possible to develop fully first-order methods given the current SimFBO framework.

**Questions:**

Another question is: In the right plot of Fig. 2, is the legend correct? It missed SimFBO.

Overall, this is a strong work with a promising solution to federated bilevel optimization, so I suggest the acceptance.


**Limitations:**

A limitation of their work is the requirement of the lower-level strong convexity in Assumption 1. It would be interesting to explore if this condition can be relaxed or eliminated, taking into account recent advancements in the field, such as those presented in [1, 2].

[1] B. Liu et al. “Bome! bilevel optimization made easy: A simple first-order approach.” NeurIPS 2022.

[2] R. Liu et al. “Averaged Method of Multipliers for Bi-Level Optimization without Lower-Level Strong Convexity.” ICML 2023.

---

> ### Author Rebuttal · Authors · 2023-08-09
>
>  We thank the reviewer TKNN for the time and valuable feedback!
>
> **Q1: Is it possible to develop fully first-order methods given the current SimFBO framework?**
>
> **A:** Good point! One possible idea is to approximate the Hessian- and Jacobian-vector products using the finite-difference tricks, i.e., $[\nabla_y g(x, y+\delta v)-\nabla_y g(x, y-\delta v)]/(2\delta)\approx \nabla^2_{yy} g(x, y)v$. Since this approximation error is controllable, it is possible to provide a convergence rate guarantee similar to that of SimFBO. We would like to leave it for our future study.
>
> **Q2: Another question is: In the right plot of Fig. 2, is the legend correct? It missed SimFBO.**
>
> **A:** Thanks! We will fix the legend in the revision.
>
> **Q3: A limitation of their work is the requirement of the lower-level strong convexity in Assumption 1. It would be interesting to explore if this condition can be relaxed or eliminated, taking into account recent advancements in the field, such as those presented in [1, 2].**
>
> **A:** Great suggestion! In this setting, it is critical but challenging to find the correct convergence criterion from the KKT perspective, develop feasible distributed constrained optimization-based methods, and provide the convergence analysis. We will leave this interesting exploration for future study.

---

> > ### Comment · Reviewer_TKNN · 2023-08-15
> >
> > Thanks for the rebuttal and I do not have further question for the moment.

---

### Official Review · Reviewer_MEss · 2023-07-07

**Soundness:** 4 excellent
**Presentation:** 4 excellent
**Contribution:** 4 excellent
**Rating:** 7
**Confidence:** 4

**Summary:**

This work studies the federated bilevel optimization, where the lower- and upper-level objectives are defined over all clients. Since the lower-level solution is the minimizer of the average of all client objectives (i.e., in a global manner), the main computational challenge is to compute the global hypergradient with second-order derivatives. This work provides a simple and effective approach named SimFBO to this problem. It contains simultaneous updates on all three variables at each client, a generalized server-level update. It has no subloops. A variant of SimFBO called as ShroFBO is also proposed to deal with the heterogeneous client computation.
Theoretical convergence is provided for these two approaches.  Experiments seem to show the proposed methods are much better than other baselines.


**Strengths:**

1. The work is well motivated and easy to follow. Finding a simple but effective solution to a complicated problem in federated setting is a interesting and important topic.

2. The algorithm design is new, i.e., simultaneous local updates, aggregation, and generalized server-side operation. The simple structure will be useful in practice. But such simple structure seems to be not easy to prove in the federated setting. Specifically, it is usually challenging to guarantee the boundedness of $v$ variable. The authors show this via induction that as long as the server-side $v$ is bounded, the local $v$ is bounded as well. This seems to be new.

3. Nonasymptotic convergence is provided and seems to be significant. It allows for client sampling without replacement, linear speedup, achieves the $1/\epsilon$ communication complexity. In contrast, existing studies require clients to be selected with replacement due to the complex hypergradient construction. This work achieves the linear speedup without replacement, and as well as optimal communication complexity. These are new results in federated/distributed optimization.

**Weaknesses:**

For the experimental part, can the authors explain why FedNest and AggITD perform poorly over CNNs? Is it due to a non-proper hyperparameter tuning or something else? More details should be provided.

**Questions:**

1. Does the $1/\epsilon$ communication complexity hold for both partial and full client participation? Per my reading, it only holds for the full participation case, right? I think the authors should make this clearer.

2. From the experiments, it is not clear to me how many local steps are used for the proposed methods? Also it would be great to investigate the impact of number of local steps on the final convergence rate and test accuracy.

3. Is it possible to incorporate the variance reduction or momentum-based estimators to improve the sample complexity to $O(\epsilon^{-3/2})$ under the current simple framework? It would be great to include some discussions on this.

**Limitations:**

yes

---

> ### Author Rebuttal · Authors · 2023-08-09
>
> We thank the reviewer MEss for the time and valuable feedback!
>
> **Q1: Why FedNest and AggITD perform poorly over CNNs?**
>
> **A:** As we illustrate in Figure 1, FedNest and AggITD both contain one or more sub-loops of communication rounds in each outer iteration, which leads to a high per-iteration computational cost as well as a much slower convergence rate. In addition, we show that the generalized server-side aggregations and updates are helpful to improve communication efficiency. However, FedNest and AggITD do not have such features. We will clarify this in the revision.
>
> **Q2: Does the **$1/\epsilon$** communication complexity hold for both partial and full client participation?**
>
> **A:** Great question! The $1/\epsilon$ communication complexity holds for full client participation. We will clarify this in the revision.
>
> **Q3: From the experiments, it is not clear to me how many local steps are used for the proposed methods. Also, it would be great to investigate the impact of the number of local steps on the final convergence rate and test accuracy.**
>
> **A:** For the experiments shown in Figure 2 and 3, we set all numbers of local steps of our SimFBO method to one because we found that more local steps only improved the performance and convergence rate marginally. In particular, we found that the performance and efficiency of SimFBO were improved only slightly when increasing the number of local steps from 1 to 5, whereas further increasing the number of steps degraded the performance due to the larger client drift. We will include such a figure in the revision.
>
> **Q4: Is it possible to incorporate the variance reduction or momentum-based estimators to improve the sample complexity under the current simple framework?**
>
> **A:** Great point! As we explained in line 120, our flexible coefficients $a_i^{(t,k)}$ of local aggregations in (4) cover the optimizers such as variance-reduced and momentum-based methods, and hence it is possible to further improve the sample complexity with these two techniques. We would like to leave this interesting work for the future.

---

> > ### Comment · Reviewer_MEss · 2023-08-16
> >
> > Thank you for the response. It clarifies my questions.

---

### Official Review · Reviewer_8pmc · 2023-07-20

**Soundness:** 3 good
**Presentation:** 1 poor
**Contribution:** 2 fair
**Rating:** 6
**Confidence:** 3

**Summary:**

The paper addresses the bilevel optimization problem in the federated learning context. The authors propose a novel gradient-based algorithm that effectively updates the arguments in both the inner and outer optimization problems simultaneously. Additionally, the paper extends this algorithm to handle data heterogeneity, making it more relevant for real-world scenarios. The theoretical analysis on the convergence rate of the proposed approach is also provided. While the content itself is interesting, the manuscript requires substantial proofreading and improvements before it is ready for peer review.

**Strengths:**

- The paper demonstrates a comprehensive treatment of the bilevel optimization problem in federated learning. Notably, the proposed method addresses data heterogeneity, contributing to the algorithm's practical applicability.

- The inclusion of a comparison with existing methods in Table 1 is valuable.

**Weaknesses:**

- There are still rooms to improve the presentation of the paper.

    - The paper suffers from unclear or inconsistent mathematical notations. E.g.

        - In defining the objective functions, a bracket should be used otherwise the meaning is not correct. Eg. in (1),
        $$\min \{\Phi(x) = F(x.y^*(x))\}$$
        The objective functions defined elsewhere in the manuscript has the same issue.

        - In (3), the notations $\zeta_i$ and $\xi_i$ are not defined. Similarly in (4). It is important to highlight they corresponds to local mini-batch dataset, as they make a difference in the proof of Lemmas in the appendix.

        - In assumption 1, it is better to use $\mu_g$ for the $\mu$-strongly convex constant as it is used elsewhere.

        - The notation $\delta_t'$ in Propostiation 2 is not defined. Though the reviewer find the definition in the appendix, the notation should be defined when it is firstly used. Similarly for other notations such as $\bar\tau$ in Theorem 1.

        - In Appendix $C$, Lemma $2$ is w.r.t. the function $G$ not $g$.


    - There are numerous typos and missing details in the appendix. E.g.

        - In Proof of Lemma 3, the statement "step 6 of algorithm (1)" does not match the algorithm description.

        - The proof of Lemma 4 missed some key steps and hence is not clear.

        - In the proof of Lemma 8 in L569. The first equality should have expectation on the LHS and the second equality should be $\leq$.

    - Presentation

        - The paper could benefit from providing concrete examples of bilevel optimization after (1) to underscore the problem's importance.

        - The main text should reference where readers can find proofs for propositions.

        - The description of the experimental setting is vague; a clearer explanation of why the studied problem is a bilevel optimization problem is needed.


- The novelty in the proposed method is limited.
    - The key method in Section 2.3 is a distributed version of the centralized gradient-based bilevel optimization method.

    - The method for hetergoeneity is a simple generalization of FedNova.

- Since the reviewer thinks the proposed method is a simple extension of existing algorithms under federated learning setting, the convergence rate analysis of the proposed algorithm is therefore a key contribution of the paper. There are several issues in technical details.

    - In Lemma 5, what is the definition of $\nabla R_i(x,y,v)$? Is it only the partial derivative w.r.t. $v$? Otherwise the gradient is not well defined since $g_i$ and $f_i$ are only assumed to be twice differentiable.

    - Due to the unclear definition of the notations and many tiny typos in the manuscript, the proof is not very easy to follow. For example, Lemma 10 is a key lemma for the proof of theorem 1, but it is not clear why the first step (24) holds since $\delta_t$ is not defined. Therefore, it is hard for the reviewer to check whether the proof is correct or not.

    - In the proof of the theorem 1, it is not clear to the reviewer how the projection of $v$ is reflected in the proof of the theorem.


    - The proof of Theorem 2, it does not make sense to the reviewer that "taking $w_i=p_i$" in L662.


- It would greatly improve the quality of the paper if the summary of the results in Table 1 can be empirically demonstrated in the experiment.


Reference:

[1] Gradient-based Bi-level Optimization for Deep Learning: A Survey

**Questions:**

See the questions in the above section.

---

> ### Author Rebuttal · Authors · 2023-08-09
>
> We thank the reviewer 8pmc for the time and valuable feedback!
>
> **Q1: Unclear notations, typos, missing details and other presentation issues.**
>
> **A:** Sorry about the confusion and missed details. Also thanks for pointing them out for us! We will definitely follow your suggestions to improve the presentation, double-check the typos and make the notations consistent.
>
> **Q2: The novelty of the proposed method is limited. The key method in Section 2.3 is a distributed version of the centralized gradient-based bilevel optimization method.**
>
> **A:** We respectfully disagree with this point. Our distributed method in Section 2.3 contains substantial new components that do not exist in the distributed version of the centralized gradient-based bilevel method. First, we include a generalized server-side aggregation and update to guarantee convergence, improve efficiency, and deal with the harder system-level heterogeneity. Second, we include a server-side projection on $v$ updates to ensure the local $v$ boundedness (via induction) and the convergence under the client drift. These key features also facilitate the design and analysis of ShroFBO under the system-level heterogeneity.
> Note that all previous federated bilevel methods do not have these new features.
>
> **Q3: What is the definition of** $\nabla R_i(x,y,v)$ **? Is it only the partial derivative w.r.t.** $v$ **?**
>
> **A:** Yes, it is the partial derivative w.r.t. $v$. We will revise the notation to $\nabla_v R_i(x,y,v)$.
>
> **Q4: Lemma 10 is a key lemma for the proof of theorem 1, but it is not clear why the first step (24) holds since is not defined.**
>
> **A:** $\delta_t$  is a positive tunable parameter that is decided later in the final convergence rate analysis, as can be seen in (30). We will clarify this in the revision.
>
> **Q5: In the proof of theorem 1, it is not clear to the reviewer how the projection of** $\delta_t$ **is reflected in the proof of the theorem.**
>
> **A:** The projection of $\delta_t$ is reflected in two places of the proof. First, we use this server-side projection to guarantee the boundedness of $v^{(t)}$ for all $t$ such that we can show via induction that the local iterates $v_i^{(t,k)}$ are bounded. Second, the projection appears in characterizing the per-iteration estimation gap $\mathbb{E}\\|v^{(t+1)} - v^*(x^{(t)})\\|^2$ in $v$ updates via the following inequality:
> $\mathbb{E}\\|v^{(t+1)} - v^*(x^{(t)})\\|^2 = \mathbb{E} \\| \mathcal{P_r}(v^{(t)} - \rho^{(t)} \gamma_v \sum_{i \in C^{(t)}} \widetilde{w_i} h_{v,i}^{(t)}) - v^*(x^{(t)}) \\|^2 \leq \mathbb{E}||v^{(t)} - v^*(x^{(t)}) - \rho^{(t)}\gamma_v\sum_{i \in C^{(t)}}\widetilde{w_i} h_{v,i}^{(t)}||^2.$
> Here, the inequality uses the non-expansive property of projection on a convex set and $v^*(x^{(t)})=\mathcal{P}_r(v^*(x^{(t)}))$ due to the boundedness of $v^*(x^{(t)})$. We will clarify this in the revision.
>
> **Q6: In the proof of Theorem 2, it does not make sense to the reviewer that "taking** $w_i = p_i$**" in L662.**
>
> **A:** Sorry for the confusion. Note that the server-side aggregations in (7) for ShroFBO replace the weights $w_i,i=1,...,n$ of the aggregations in (6) for SimFBO by $p_i,i=1,...,n$. It is equivalent to setting $w_i$ to be $p_i$ in the analysis of Theorem 1 for SimFBO. We will clarify this and directly use $p_i$ in the proof of the Theorem in the revision.

---

> > ### Comment · Reviewer_8pmc · 2023-08-13
> >
> > Thanks for the rebuttal and the clarification on the misunderstanding that I might have before. I have updated my scores based on the rebuttal.

---

> > > ### Author Response · Authors · 2023-08-13
> > > **Thanks so much for your updates**
> > >
> > > Dear Reviewer,
> > >
> > > Thanks so much for your updates and for raising your score. We are happy that our responses clarify your questions. We will take your suggestions into our revision.
> > >
> > > Best,
> > > Authors

---

### Decision · Program_Chairs · 2023-09-21

**Decision:**

Accept (spotlight)

**Comment:**

All the reviewers are consistently positive about this paper and agree that this paper is worthwhile to be accepted. Most of the concerns seem to be clarified during the rebuttal, and I would recommend accepting this paper.